# Co-Expression Network Analysis Unveiled lncRNA-mRNA Links Correlated to Epidermal Growth Factor Receptor-Tyrosine Kinase Inhibitor Resistance and/or Intermediate Epithelial-to-Mesenchymal Transition Phenotypes in a Human Non-Small Cell Lung Cancer Cellular Model System

**DOI:** 10.3390/ijms25073863

**Published:** 2024-03-29

**Authors:** Valentina Fustaino, Giuliana Papoff, Francesca Ruberti, Giovina Ruberti

**Affiliations:** Institute of Biochemistry and Cell Biology, National Research Council (IBBC-CNR), Campus Adriano Buzzati Traverso, Via E. Ramarini 32, 00015 Monterotondo (Roma), Italy; giuliana.papoff@cnr.it (G.P.); francesca.ruberti@cnr.it (F.R.)

**Keywords:** NSCLC, WGCNA, lncRNA-mRNA networks, EGFR-TKI resistance, intermediate EMT phenotypes

## Abstract

We investigated mRNA-lncRNA co-expression patterns in a cellular model system of non-small cell lung cancer (NSCLC) sensitive and resistant to the epithelial growth factor receptor (EGFR) tyrosine kinase inhibitors (TKIs) erlotinib/gefitinib. The aim of this study was to unveil insights into the complex mechanisms of NSCLC targeted therapy resistance and epithelial-to-mesenchymal transition (EMT). Genome-wide RNA expression was quantified for weighted gene co-expression network analysis (WGCNA) to correlate the expression levels of mRNAs and lncRNAs. Functional enrichment analysis and identification of lncRNAs were conducted on modules associated with the EGFR-TKI response and/or intermediate EMT phenotypes. We constructed lncRNA-mRNA co-expression networks and identified key modules and their enriched biological functions. Processes enriched in the selected modules included RHO (A, B, C) GTPase and regulatory signaling pathways, apoptosis, inflammatory and interleukin signaling pathways, cell adhesion, cell migration, cell and extracellular matrix organization, metabolism, and lipid metabolism. Interestingly, several lncRNAs, already shown to be dysregulated in cancer, are connected to a small number of mRNAs, and several lncRNAs are interlinked with each other in the co-expression network.

## 1. Introduction

Lung cancer is a leading cause of cancer-related deaths worldwide, with extremely high morbidity and mortality rates. NSCLC is the most common type of lung cancer, and it represents 85% of all cases [1,2].

Increasing evidence highlights the important role of non-coding RNA (ncRNA) in regulating the tumorigenesis process by modulating crucial signaling pathways, gene expression of proto-oncogenes or tumor suppressors. Non-coding sequences covering 98% of the human genome are divided into different classes based on their length, localization, and/or function. MicroRNA (miRNA), long non-coding RNA (lncRNA), and circular RNA (circRNA) are either up- or downregulated in cancer types and can promote or suppress the progression of the disease. LncRNAs, arbitrarily defined as non-coding transcripts of more than 200 nucleotides (200 nt) according to the recent classification and nomenclature, are RNA mostly generated by RNA Polymerase II [3]. LncRNAs can be grouped according to their position relative to the protein-coding genes. As a result, they can be roughly divided into antisense, enhancer, bidirectional (divergent), intronic transcript lncRNAs, and large intergenic non-coding RNAs. LncRNAs may regulate gene expression via multiple mechanisms at the epigenetic, transcriptional, and post-transcriptional levels [4]. LncRNAs have been shown to participate in the regulation of a variety of cell activities through interaction with other RNAs, DNAs, or proteins, including cell differentiation, proliferation, invasion, apoptosis, and autophagy. Cooperation among lncRNAs can further amplify the role of lncRNAs in physiological and pathological processes [5,6,7]. Transcriptional dysregulation of lncRNAs has been correlated to cancer growth, metastasis, and drug resistance in NSCLC [8,9,10,11,12,13,14,15]. TKIs targeting EGFR are the primary treatment for NSCLC harboring activating mutations in the EGFR TK domain; unfortunately, resistance to EGFR-TKIs is unavoidable, and most of the patients experience a rapid cancer relapse [16,17,18].

The association between EMT plasticity and drug resistance is very well documented by many in silico, in vitro, and in vivo studies [19,20]. The EMT is a cellular process in which polarized epithelial cells undergo multiple molecular and biochemical changes and lose their identity to acquire a mesenchymal phenotype. Importantly, EMT is a complex, dynamic, and multifaceted process characterized by a spectrum of intermediate phenotypes whose molecular hallmarks remain poorly characterized and cells can be stably arrested in an intermediate, or hybrid, state, contributing to cancer collective cell migration, and cell cluster formation and dissemination, associated with enhanced tumor aggressiveness and worse clinical outcomes [21,22]. Intermediate epithelial/mesenchymal phenotypes have also been described in lung, breast, prostate, ovarian, pancreatic, and many other cancer types by our research group [23,24,25,26,27,28,29,30]. The intrinsic complexity of the process of EMT transformation in cancer epithelial cells and the influence of genomic and microenvironmental elements in its shaping are emerging from bioinformatics and mathematical modeling studies [31,32].

NcRNAs regulate the EMT at multiple levels, including gene regulation of transcription factors, cell adhesion, cytoskeleton organization, and cell motility signaling pathways. The impact of ncRNAs, including lncRNA, is still far from being fully understood. Although lncRNAs are usually described as EMT promoters or EMT-suppressors, some of them have controversial functions in different types of tumors or different conditions, underlying the complexity and plasticity of tumor cells [33,34]. Furthermore, the contribution of lncRNAs to intermediate EMT phenotypes is still unknown.

Many studies consider only differences in the expression of genes between different samples, ignoring the underlying connection of each gene. Weighted gene co-expression network analysis (WGCNA) is a systematic bioinformatics method used to describe correlation patterns among genes in samples and can identify clusters of highly correlated genes (hereafter *modules*). This approach also explores the relationship between modules and phenotypes of interest [35].

In this study, we performed a WGNA on a microarray gene expression dataset (GSE80344) publicly available at the NCBI’s Gene Expression Omnibus (GEO) database and previously reported by us [30]. The dataset was obtained from eight human NSCLC cancer cell lines: two EGFR-TKI (erlotinib, gefitinib)-sensitive cell lines and six derived EGFR-TKI-resistant cell lines. We aimed to identify lncRNA–mRNA sub-networks associated with resistance to EGFR-TKIs and/or intermediate EMT phenotypes. Our analysis led to the selection of three modules highly correlated with coding genes involved in key biological pathways and processes, including Rho (A, B, C) GTPase and/or Rho GTPase regulation, apoptosis, positive regulation of I-kappaB kinase/NF-kappaB signaling, NLRP3 inflammasome complex assembly, cell adhesion, cell migration, cell–extracellular matrix interactions, metabolism, lipid metabolism, interleukin (4, 12, and 13), phosphatidylinositol, and RAS signaling pathways, and signaling pathways associated with cancer. Interestingly, several lncRNAs are highly connected to genes belonging to these pathways. This analysis may offer novel insights into the functional study of unknown lncRNAs correlated to EGFR-TKI resistance and intermediate EMT phenotypes to reach a better understanding of the molecular pathways contributing to these phenotypes.

## 2. Results

### 2.1. Construction of a Weighted Gene Co-Expression Network

A weighted co-expression network of the transcriptome of our NSCLC cellular model system was built with the objective of defining modules related to phenotypes of interest and, among those, selecting the most relevant lncRNAs. An unsigned WGCNA was performed in *R* complying with the pipeline defined by Peter Langfelder and Steve Horvath to identify genes both positively and negatively correlated with the phenotype traits (Figure 1A) [35]. Briefly, we used the Pearson correlation to calculate the co-expression of probes and a power adjacency function to build the network by determining a soft threshold *β* = 7, based on the criterion of approximate scale-free topology (Figure 1B). Using the Dynamic Tree Cut and Merged Dynamic algorithms, the gene probes were grouped into modules according to their topological overlap matrix (TOM) scores (Figure 1C; see Methods for details). The assignment of probes to each module, as well as their connectivity measures and module memberships, are reported in Appendix A.

At the end of this step, we had a WGCNA network of the transcriptome of the NSCLC cell lines, where each connection was weighted, and this weight allowed the subdivision of the network into 47 modules of highly correlated genes.

### 2.2. Definition of Module–Trait Relationships and Detection of Key Modules

The modules identified by WGCNA were then correlated with the phenotypic traits of interest: EGFR-TKI resistance (ERL-res) and intermediate EMT phenotypes (Figure 2). For the definition of the intermediate EMT phenotypes, we used two comparisons: “*I* vs. *E*”, to select modules with gene expression differences between cell lines with intermediate EMT phenotypes (*I*) and epithelial cell lines (*E*); and “*I* vs. *M*”, to select modules with gene expression differences between cell lines with intermediate EMT phenotypes and mesenchymal cell lines (*M*). We used a binary code for all phenotypic traits as suggested by the WGCNA package manual [35] (Appendix A). To identify genes associated with both intermediate EMT phenotypes and EGFR-TKI resistance, we focused our attention on modules with: (a) high significant correlation with ERL-res and “*I* vs. *E*” traits (|*ρ_s_*| > 0.65, *p*-value < 0.05); (b) significant correlation with the “*I* vs. *M*” trait (*p*-value < 0.05). From these analyses, we identified modules positively or negatively correlated to the phenotypic traits, although upregulated or downregulated genes are both present in each module (Appendix A).

To avoid biased results dictated by the difference in cellular origins of the NSCLC model system, we discarded from the selection all the modules (n = 19) that had a significant correlation with the cell lineage trait.

Based on this rationale, we selected three modules: *brown4* (Appendix A), deeppink (Appendix A), and *magenta4* (Appendix A). We also selected the module *plum4* (Appendix A) because of its high and significant correlation with ERL-resistance (*ρ_s_* = 0.75, *p*-value < 0.05) and its enrichment in probes specific for lncRNAs. Next, to verify that connectivity and memberships had good correlations with our phenotypic trait of interest, we performed a correlation analysis using the absolute values of module membership and gene significance as well as gene significance and intramodular connectivity of probes (Appendix A). Data regarding positive and negative membership and the gene significance of the individual members of modules are listed in Appendix A. While *brown4*, *magenta4*, and *plum4* modules resulted in good significant correlations between gene significance and module membership and/or connectivity (*ρ_s_* > 0.3, *p*-value < 0.05), the *deeppink* module showed unacceptable parameters; therefore, it was not further investigated (Appendix A).

To summarize, among the 47 modules individuated by the WGCNA, we selected 3 modules: *brown4*, *magenta4*, and *plum4*. They have a high significant global correlation with ERL-resistance and/or intermediate EMT phenotypes; and good significant correlations between the gene significance of these phenotypic traits and the centrality measures of module members (Figure 2C). 

### 2.3. Functional Enrichment Analysis of the Brown4 and Magenta4 Modules

We then investigated the biological pathways affecting protein coding genes in the *brown4* and *magenta4* modules. We performed a functional enrichment analysis using the biological processes and pathway terms of the DAVID database. The results of the top enriched pathways (fold-enrichment > 2) are presented in Figure 3, while all significant enriched terms are reported in Appendix A. In the brown4 module, the genes were mainly enriched in seven Gene Ontology biological processes (GO-BP) and in five Reactome pathways, including Rho GTPases, Rho GTPase regulation, *apoptosis*, *positive regulation of I-kappaB kinase/NF-kappaB signaling and positive regulation of NLRP3 inflammasome complex assembly*, *iron ion homeostasis, or iron–sulfur cluster* (Figure 3). 

*Magenta4* shows enrichment in several pathways: 3 KEGG, 8 GO-BP, and 14 Reactome pathways. Remarkably, a good portion of enriched pathways belong to *cell adhesion*, *cell migration*, *cell–extracellular matrix interactions*, *metabolism*, *and/or lipid and phospholipid metabolism terms*, *interleukin signaling pathways*, particularly 4, 12, and 13, *Phosphatidylinositol*, and *RAS pathways* (Figure 3). 

It was not possible to obtain an accurate functional enrichment analysis in *plum4*, as it has a small size and a high portion of lncRNAs. Despite this, it shows a slight enrichment in the *negative regulation of transport* GO-BP term (Appendix A).

Taken together, the functional enrichment analysis identified about 37 key biological processes and signaling pathways that could be involved in ERL-resistance and/or intermediate EMT-phenotypes. Among these, terms regarding Rho GTPase signaling, apoptosis, cell migration, lipid metabolism and cancer-related signaling pathways, seem to be recurrent between and within the consulted databases.

### 2.4. Selection of the Most Relevant LncRNAs

To identify the lncRNAs with the best correlation in the module network, as well as the phenotypic traits of interest, we selected the lncRNAs with high significative scoring of module membership (|MM| > 0.7, *p*-value < 0.05) and gene significance (|GS| > 0.7, *p*-value < 0.05). A summary of the best lncRNAs of the *brown4*, *magenta4*, and *plum4* modules, ranked according to their intramodular connectivity (Ki), is listed in Table 1 and Table 2. Within each module, several biotypes were found, including long intergenic non-coding RNA (lincRNA), miRNA host genes, divergent lncRNAs, and antisense lncRNAs. A relevant number of lncRNAs were present in the modules: *brown4*—20 lncRNAs (23 probes mapping to 20 genes); *magenta4*—26 lncRNAs (29 probes mapping to 26 genes); and *plum4*—8 lncRNAs (9 probes mapping to 8 genes) (Table 1, Table 2, and Appendix A).

### 2.5. Expression Validation of Selected LncRNAs

The expression of some lncRNAs, differentially expressed in the erlotinib-sensitive and -resistant NSCLC cell lines in the microarray data and belonging to the three selected WGCNA modules, was validated by RT-qPCR. RNA expression was analyzed after verification of sensitivity to erlotinib, as described in the Material and Methods section, by MTT viability assays. As previously described, both HCC827 and HCC4006 cell lines are highly sensitive to erlotinib targeting the EGFR, while their derived cell lines (i.e., RA1, RA2, RB1, RB1.1, and RB2 derived from HCC827 and RC2.2 derived from HCC4006) are stably resistant to erlotinib (IC_50_ > 10 μM).

LINC-PINT, LYPLAL1-DT, and MIR100HG of the *brown4* module; GIHCG, ZMIZ1-AS1 of the *magenta4* module; and SATB2-AS1 of the *plum4* module are all upregulated in the erlotinib-resistant cell lines, while MIR205HG of the *magenta4* module is downregulated (Figure 4). These data agree with the microarray expression profile previously reported [30]. The direction of the correlation of lncRNAs with the phenotypic traits of interest, as well as the direction of their correlation with modules, are listed in Appendix A.

### 2.6. Analysis of mRNA-lncRNA Sub-Networks

Co-expression networks were built for the three selected modules, as described in methods, to investigate the best connected lncRNA–mRNA genes and, in turn, to study the involved biological processes and signaling pathways of module functional enrichment analysis (Figure 3, Appendix A). It is worth mentioning that, because our data are at probe-level, each gene in the network could be represented by multiple nodes, one for each probe.

The network of the *brown4* module contains 87 mRNAs (93 probes) and 29 lncRNAs (34 probes), 20 of them represent the best lncRNAs as highlighted by our selection criteria, mentioned above and in the Methods section (Table 1 and Appendix A, Appendix A). Interestingly, 14/20 lncRNA nodes (C2CD4D-AS, ENSG00000261490, ENSG00000287839, GABPB1-AS1, GAPLINC, LINC-PINT, LINC00653, LINC01004, LINC01547, LOC107984035, LYPLAL1-DT, MIR100HG, PAN3-AS1, ZNF436-AS1) show connectivity with 5 mRNAs (ARHGEF10L, BCR, PCDH7, SFN, SLC4A7) belonging to *Rho GTPase* and/or *Rho GTPase regulation* enriched pathways (Figure 5A, Appendix A). As shown in Figure 5A, multiple probes for the lncRNA LINC-PINT, all mapping to the same transcripts, are in the subnetwork. Moreover, 8/20 lncRNAs (ENSG00000261490, GAPLINC, LINC-PINT, LINC01547, LOC107984035, LYPLAL1-DT, PAN3-AS1, ZNF436-AS1) show connectivity with 4 mRNAs (CIDEC, PDCD6IP, PSMC2, SFN) belonging to apoptotic enriched processes and pathways (Figure 5B, Appendix A). It is to be considered that some lncRNAs of the two sub-networks are in common (ENSG00000261490, GAPLINC, LINC-PINT, LINC01547, LOC107984035, LYPLAL1-DT, PAN3-AS1, ZNF436-AS1), highlighting the link between Rho GTPase and apoptotic pathways. The lncRNAs LINC-PINT, LOC107984035, MIR100HG, PAN3-AS1 were also connected to mRNAs (APOL3, CARD11, MYD88, SLC44A2) of the enriched Gene Ontology term *positive regulation of I-kappaB kinase/NF-kappaB signaling* and *positive regulation of NLRP3 inflammasome complex assembly* (Appendix A). None of the selected lncRNAs were connected to the genes of the *iron ion homeostasis* or *iron-sulfur cluster assembly* enriched pathways.

The network of the *magenta4* module contains 212 mRNAs (231 probes) and 34 lncRNAs (37 probes), 26 of them represent the best lncRNAs selected (Table 1, Appendix A, and Appendix A). Interestingly, 15/26 lncRNA nodes (ENSG00000233085, ENSG00000236453, ENSG00000251532, ENSG00000268403, ENSG00000289039, FAM95C, GIHCG, LASTR, LINC01182, MAP3K2-DT, MHENCR, MIR205HG, NEAT1, RAB30-DT, ZNF674-AS1) show connectivity with 11 mRNAs (CEACAM1, CLSTN1, COL4A6, FGF2, IGSF9, ITPR2, NDNF, RASGEF1A, SPTB, USP9X, VAV1) belonging to cell adhesion, cell migration, and cell–extracellular matrix interactions (Appendix A, Figure 6). Moreover, 24/26 lncRNAs (ENSG00000233085, ENSG00000236453, ENSG00000251532, ENSG00000257283, ENSG00000267575, ENSG00000268403, ENSG00000289039, FAM95C, GIHCG, KDM7A-DT, LASTR, LINC00173, LINC00472, LINC00973, LINC01182, LINC01629, LINC02512, LOC100506124, MAP3K2-DT, MHENCR, MIR205HG, RAB30-DT, ZMIZ1-AS1, ZNF674-AS1) show connectivity with 19 mRNAs (ACSL4, CKMT1B, FAR1, FDFT1, GCH1, GLS2, IMPA2, ITPR2, LGMN, LRP12, MGLL, OAZ3, PLA2G10, PLAAT2, RPL3, SLC27A3, SYNJ2, TBXAS1, TPK1) belonging to *metabolism*, and/or *lipid and phospholipid metabolism* terms (Appendix A, Figure 6). In addition, 7/26 lncRNAs (ENSG00000257283, ENSG00000289039, GIHCG, LASTR, LINC00173, LINC00973, MHENCR) show connectivity with 5 mRNAs (FGF2, MUC1, STAT4, VAV1, ZEB1) belonging to interleukin signaling pathways, in particular 4, 12, and 13. Finally, 13/26 lncRNAs (ENSG00000233085, ENSG00000236453, ENSG00000268403, ENSG00000289039, FAM95C, GIHCG, LASTR, LINC00173, LINC01182, LINC02512, MAP3K2-DT, MHENCR, ZNF674-AS1) show connectivity with 8 mRNAs (COL4A6, FGF2, IMPA2, ITPR2, LPAR2, RASGEF1A, STAT4, SYNJ2) belonging to Phosphatidylinositol, RAS, and signaling pathways associated with cancer (Appendix A, Appendix A).

The network of *plum4*, a small module characterized by many lncRNAs, contains 9 mRNAs (9 probes) and 14 lncRNAs (15 probes), of which 8 represent the best lncRNAs selected (Table 2 and Appendix A, Figure 7). Five out of eight lncRNA nodes (DPY19L3-DT, ENSG00000288989, LINC01358, LINC02814, SATB2-AS1) show connectivity with a single mRNA coding for the mitochondrial inner membrane protein-like protein, MPV17L, predicted to be involved in several processes, including cellular response to reactive oxygen species and apoptosis.

Interestingly, the presence of several interconnected lncRNAs in the *brown4* and *plum4* modules offers the possibility of investigating the effects of lncRNA cooperation in cell signaling pathways and NSCLC-associated phenotypes.

## 3. Discussion

NSCLC is the most widespread of all lung cancers and a leading cause of cancer-related mortality worldwide. TKIs targeting the EGFR protein serve as a critical pillar in the treatment of NSCLC, but unfortunately, resistance is unavoidable. Identifying the potential key factors of drug resistance to EGFR-TKIs is essential to treating patients with EGFR mutant lung cancer and to developing novel therapeutic strategies. LncRNAs act as versatile regulators involved in diverse biological processes in cancer, including drug resistance [8,9,10,11,12,13,14,15] and EMT [33,34]. However, their contribution to intermediate EMT phenotypes is still unknown.

LncRNAs are generally expressed at lower levels than protein-coding RNAs; therefore, they might be overlooked by conventional pair-wise gene expression comparisons [36]. However, despite their low abundance, they can exert pronounced effects via their interplay with other nucleic acid and protein molecules.

WGCNA has many advantages over other differential expression analysis methods since its focus is on co-expression patterns, which helps discover functional modules containing related genes. Therefore, we decided to use WGCNA to interrogate the gene expression data of erlotinib-sensitive and -resistant NSCLC cell lines that we isolated in the lab, characterized also by an intermediate EMT phenotype [30,37], to identify lncRNA–mRNA links.

In this study, a total of 20,192 probes (mapping to 13,904 Ensembl gene ID) were used to carry out WGCNA, and 47 modules with sizes ranging from 32 to 4725 probes per module were generated. The three most significant modules, *brown4*, *magenta4* and *plum4*, were selected for further analysis. Several critical and interesting biological processes, lncRNAs and mRNAs were identified.

Intriguingly, the *brown4* module shows 14 out of 20 selected lncRNA nodes connected with 5 mRNAs, belonging to Rho GTPase and/or Rho GTPase regulation enriched pathways.

The role of Rho GTPases, as key regulators of biological processes relevant for cancer development and progression, has been investigated for decades [38,39]. The Rho GTPases, a family of highly conserved GTPases that are encoded by 20 genes in humans, regulate a range of cellular functions. They are activated via the dysregulation of expression and/or activity of a myriad of oncogenic cell surface receptors, including growth factor receptors. In turn, Rho GTPases signal to multiple downstream effectors that regulate migration/invasion via de novo actin polymerization, cell polarization, metastasis, EMT, cell proliferation, cell cycle progression, apoptosis/survival, vesicle trafficking, angiogenesis, cell–cell, and cell-substrate adhesions.

The interaction of the Rho GTPases with several targets, the signals originated from distinct subcellular pools of Rho GTPases, and their spatial and temporally regulation can drive diverse physiological and pathological outcomes [39]. The impact of lncRNAs on Rho GTPase signaling has been shown to be exerted through direct modulation of their expression, by influencing the expression of miRNAs that negatively regulate Rho GTPases, or by acting as molecular sponges of relevant miRNAs [40,41]. Interestingly, among the lncRNAs and connected mRNAs of the *brown4* network (Appendix A) belonging to the enriched Rho GTPase pathway, there is a bicarbonate transporter (SLCA47), already shown to be dysregulated in epithelial cancers, and several linked lncRNAs (C2CD4D-AS, ENSG00000261490, ENSG00000287839, GABPB1-AS1, GAPLINC, LINC-PINT, LINC00653, LINC01004, LOC107984035, MIR100HG, PAN3-AS1) (Figure 5).

Within the last decade, numerous studies have demonstrated that intracellular pH homeostasis is often dramatically altered in cancer. Because of the vital importance of the acid–base balance, all living cells have systems to maintain the stability of their intracellular and extracellular pH (pHi and pHe). Evidence has emerged that the high metabolic requirement of proliferating cancer cells, the shift toward a glycolytic metabolism as a response to hypoxia, as well as oncogene-driven changes in gene expression, lead in cancer tissues to an increase in pHi compared to normal cells with a pHe decrease, hence a reversal of pHi < pHe of normal tissue cells. Therefore, the dysregulation of the expression and activity of pH-regulatory proteins, H^+^ and HCO^3−^ transporters are areas of intense study in cancer [42,43,44,45,46,47].

The Na^+^-HCO^3−^ co-transporter SLC4A7 (also known as NBCn1), a key contributor to epithelial pH homeostasis, is upregulated at the mRNA level in our erlotinib-resistant cell lines. Interestingly, it is heavily connected to several lncRNAs in the *brown4* network. SLC4A7 is also a target of mTORC1 signaling and sustains the mTORC1-dependent control of nucleotide synthesis [48,49]. Moreover, SLC4A7 has also been reported to promote EMT and metastasis of the head and neck squamous cell carcinoma; similarly, pHi dynamics studies as well as mRNA and protein expression of acid-base transporters among breast cancer patients showed that increased SLC4A7 expression predicts proliferative activity and metastasis [50,51].

Another gene worthy of mention is BCR, which is connected to lncRNAs in the *brown4* module (Figure 5) and shows a downregulation at the mRNA level in the erlotinib-resistant cell lines. Indeed, Bcr protein may have opposing regulatory activities toward small GTP-binding proteins because the C-terminus of Bcr is a GTPase-activating protein (GAP) domain, which stimulates GTP hydrolysis by RAC1, RAC2, and CDC42 [52,53,54]. Instead, the central Dbl homology (DH) domain functions as a guanine nucleotide exchange factor (GEF) that modulates the GTPases CDC42, RHOA, and RAC1 by promoting their conversion from the GDP-bound to the GTP-bound form [52,55].

Functionally, Bcr may act as an important negative regulator of Rac1 activity in different cell types [53]. Rac1, a widely expressed Rho GTPase, is a major player in the assembly of actin-rich membrane protrusions (i.e., ruffles) implicated in cancer cell migration [56,57]. Growth factors and other extracellular stimuli activate Rac1 via GEFs, and active GTP-bound Rac1 subsequently propagates motility signals via downstream effectors [58,59]. Recently, Rac-GEFs responsible for Rac1-mediated lung cancer cell migration upon EGFR and c-Met activation have been identified [60]. Whether BCR downregulation leads to Rac1 activation in lung adenocarcinoma has not yet been investigated.

In the *brown4* module, LINC-PINT, LYPLAL1-DT, and MIR100HG are among the most significant lncRNAs connected to the Rho enriched pathways; they show good values of membership (MM = 0.9), connectivity (kWithin scaled ≥ 0.6), and gene significance for ERL-resistance (GS ≥ 0.6) and EMT intermediate phenotype (GS ≥ 0.7).

The long intergenic non-protein-coding RNA, p53-induced transcript (LINC-PINT) contributes to a variety of biological processes impacting cancer cell growth and metastasis, with involvement in processes ranging from DNA damage responses to cell senescence and apoptosis [61].

The lysophospholipase-like 1-divergent transcript (LYPLAL1-DT) has been shown to act as a small cell lung cancer (SCLC) oncogenic lncRNA by in vitro and in vivo studies, promoting cell proliferation, migration, and invasion [62].

MIR100HG can regulate cell proliferation, apoptosis, cell cycle transition, and cell differentiation. It has been functionally related to several signaling pathways, such as TGF-β, Wnt, Hippo, and ERK/MAPK. Dysregulation of MIR100HG has been detected in a diversity of cancers [63,64].

These lncRNAs, already shown to be dysregulated in epithelial cancers, have not yet been investigated in these specific NSCLC cancer phenotypes or in the Rho signaling pathways; therefore, further investigation is required.

In the larger *magenta4* module, the co-expression lncRNA-mRNA network analysis highlighted two main enriched functional pathways: one related to cell adhesion, cell migration, and cell–extracellular matrix interactions, and the other belonging to metabolism and/or lipid metabolism (Figure 6).

In the first functional pathway, 15 of the 26 best selected lncRNA nodes show connectivity with 11 mRNAs. Among the lncRNAs, GIHCG and MIR205HG show very good parameters of module membership (MM = −0.9 and 0.9, respectively) and gene significance for both ERL-resistance and EMT intermediate phenotypes (GS = 0.8 and −0.8, respectively), and good connectivity measures (kWithin scaled = 0.6 and 0.5, respectively).

GIHCG was found to be upregulated in several cancer types, including hepatocellular carcinoma and cervical and renal carcinoma, and showed a role in proliferation and cell migration regulation [65,66,67].

Interestingly, using the baseline gene-expression data and corresponding drug response data from two large cell line screens (Genomics of Drug Sensitivity in Cancer, and Cancer Therapeutics Response Portal), Nath et al. found that MIR205HG can be considered a biomarker of erlotinib response in lung cancer cells [68].

Among others, FGF2 and USP9X mRNA, both upregulated in our resistant cell lines, are present in this network. The X-linked ubiquitin-specific peptidase 9 (USP9X) is a member of the deubiquitinase family shown to be significantly increased in several tumors, including non-small cell lung cancer [69]. Overexpression of USP9X in cancer activates multiple important pathways, including the PI3K/AKT, Rho/Rho-associated protein kinase, Notch, NK-κB, and Wnt/β-catenin pathways [70]. Autocrine signaling of fibroblast growth factors (FGF) and their receptors (FGFR) has been shown in NSCLC cell lines [71,72], and activation of an FGF2–FGFR1 autocrine loop has been reported in EGFR-TKI-resistant cell lines [73].

Moreover, 24 out of 26 lncRNAs show connectivity with 19 mRNAs involved in metabolism and lipid metabolism pathways. Among the lncRNA are GIHCG (discussed in the previous paragraph) and ZMIZ1-AS1 that show a MM—0.8; kWithin scaled equal to 0.5 and a good gene significance for ERL-resistance (GS = 0.8) and EMT intermediate phenotype (GS = 0.8).

The Zinc Finger MIZ-Type Containing 1-Antisense RNA 1, ZMIZ1-AS1, has been previously shown to be upregulated in an erlotinib-resistant cell line derived from HCC827 and by RNA pull-down assay and mass spectrometry to interact with the nuclear ribonucleoprotein hnRNPA1 [74].

It is very well known that tumor initiation and progression require the metabolic reprogramming of cancer cells through the regulation of the expression and activity of enzymes and transporters. Cancer cells should adapt to the increased energy demand, nutrient availability, and oxidative stress associated with cancer cell proliferation [75]. The epithelial-to-mesenchymal transition has also been linked with complex metabolic reprogramming [76,77]. Among the metabolic alterations observed in lung cancer, those associated with lipid metabolism have recently received increasing attention [78,79].

The lncRNA-mRNA network of the *magenta4* module correlated to metabolic pathways includes, among others, some interesting genes: ACSL4, FDFT1, MGLL, and GLS2. The ACSL4 gene, upregulated in the erlotinib-resistant cell lines, is a member of the long-chain fatty acyl CoA synthetase (ACSLs) family of enzymes that contributes significantly to lipid metabolism, playing a role in both anabolic (fatty acid synthesis and lipogenesis) and catabolic pathways (lipolysis and fatty acid β-oxidation). ACSLs affect the behavior of malignant cells, including proliferation, migration, invasion, apoptosis, and drug resistance. ACSL4 has also been involved in ferroptosis, a cell death pathway caused by excessive accumulation and failure to eliminate iron-dependent lethal toxic lipid reactive oxygen species (ROS) [80]. A cholesterogenic gene, Farnesyl-Diphosphate Farnesyltransferase 1 (FDFT1), downregulated in the erlotinib-resistant cell lines, is a gene encoding a membrane-associated enzyme located at a branch point in the mevalonate pathway. It is the first specific enzyme in cholesterol biosynthesis, and its relevance to tumor progression and tumor environment has been recently reviewed [81].

Monoacylglycerol lipase (MGLL, MGL, or MAGL) is an enzyme belonging to the family of serine hydrolases that preferentially catalyzes the hydrolysis of mono-triglycerides to glycerol and fatty acids. This enzyme is also the most important degrading enzyme for the endocannabinoid 2-arachidonoylglycerol (2-AG). MGLL inhibitors have been considered important agents in many diseases for their anti-nociceptive, anxiolytic, anti-inflammatory, and anti-cancer properties [82]. Interestingly, this gene is downregulated in the erlotinib-resistant cell lines and gene knock-out in mice leads to an increased incidence of lung cancer and the activation of EGFR and ERK [83].

The GLS2 gene, downregulated in the erlotinib-resistant cell lines, plays an important role in the regulation of glutamine catabolism. GLS2 protein promotes mitochondrial respiration and increases ATP generation in cells by catalyzing the synthesis of glutamate and alpha-ketoglutarate. The GSL2 oncogene or tumor suppressor gene in different cancer types has also been correlated with EMT [84,85].

The network of the small *plum4* module (Figure 7) is characterized by the presence of many lncRNAs, including SATB2-AS1, special AT-rich sequence binding protein 2 antisense RNA 1 (MM = −0.9; kWithin scaled = 0.6; GS_ERL-resistance_ and GS_EMT intermediate phentotype_ = 0.8), which has been reported to promote tumor cell growth in osteosarcoma [86], and NSCLC [87]; in contrast, it inhibits tumor cell metastasis in colorectal cancer through positive regulation of SATB2 gene expression [88]. Accordingly, in our erlotinib-resistant NSCLC cell lines, both SATB2-AS1 (*plum4* module) and SATB2 (*deeppink* module) are upregulated, suggesting a similar positive regulation of the coding gene by the lncRNA.

## 4. Material and Methods

### 4.1. Reagents

Chemicals and molecular biology reagents, if not otherwise stated, were purchased from Merck Life Science, Milan, Italy or Thermo Fisher Scientific, Milan, Italy; Erlotinib Hydrochloride Salt was purchased from LC Laboratories, Woburn, MA, USA; and MTT, 3-(4,5-methylthiazol-2-yl)-2,5-diphenyltetrazolium bromide, was purchased from Sigma-Aldrich. MTT stock solution (5 mg/mL in H_2_O, sterilized by filtration) was stored at 4 °C for 1 month. Applied Biosystem PowerUp SYBR Green Master Mix for qPCR was from Thermo Fisher Scientific, Milan, Italy. The Reverse Transcription System was from Promega Italia Srl, Milan, Italy.

### 4.2. Cell Culture

For the model of NSCLC, the study used EGFR-TKI (erlotinib/gefitinib)-sensitive and -resistant cell lines derived and characterized in our laboratory and previously described [30,37]. Our model system is composed of two parental EGFR-mutated cell lines that are sensitive to EGFR-TKI, HCC827 (ATCC CRL-2868) and HCC4006 (ATCC CRL-2871), and six EGFR-TKI-resistant cell lines: five HCC827-derived cell lines (RA1, RA1, RA2, RB1, RB1.1, and RB2), and one HCC4006-derived cell line (RC2.2). Concisely, NSCLC cells were maintained in RPMI 1640 medium (BioWhittaker, Lonza, Euroclone, Milan, Italy) supplemented with 10 mM Hepes pH 6.98–7.30, 1 mM L-glutamine, 100 U/mL penicillin/streptomycin (BioWhittaker, Lonza), and heat-inactivated 10% fetal bovine serum (FBS) (Merck Life Science, Milan, Italy). All cells were cultured at 37 °C in a 5% CO_2_ humidified incubator.

The MTT assay was performed as previously described [37]. Briefly, cells (10–20 × 10^4^ cells/well) in a 96-well plate were treated with an increasing concentration of erlotinib (from 64 pM up to 10–20 μM) in complete tissue culture medium for 72 h. Next, cells were washed with PBS, incubated for 4 h with MTT (1 mg/mL), and processed for color detection upon solubilization with DMSO. The samples were quantified spectrophotometrically at 570 nm, with a reference wavelength of 630 nm, using a Varioskan Lux multimode microplate reader (Thermofisher Scientific, Waltham, MA, USA). Data analysis was achieved using GraphPad Prism v 7.0 software (GraphPad Software Inc., La Jolla, CA, USA), and IC_50_ was obtained after nonlinear regression curve fitting, according to log (inhibitor) versus normalized response with a variable slope curve model.

### 4.3. RNA Extraction and Real-Time qPCR Analysis

Total RNA was extracted from the erlotinib-sensitive and -resistant cell lines using the total RNA purification plus kit (Norgen Biotek Corporation, Thorold, ON, Canada) and retro-transcribed with the GoScript Reverse Transcription System (Promega Italia, Srl, Milan, Italy) using random primers. Quantitative real time PCR (qRT-PCR) analysis was performed in an Applied Biosystem 7500 Fast Real-Time PCR System (from Thermo Fisher Scientific, Milan, Italy) using Applied Biosystem PowerUp SYBR Green Master Mix for qPCR. Sequences of specific primers are shown in Appendix A. Ribosomal protein L31 (RPL31) was used as a reference gene to normalize the quantitation of target genes for differences in the amount of total RNA in each sample. The relative fold change of target genes in resistant cell lines in comparison to HCC827 and HCC4006 parental cell lines was calculated using the 2^−ΔΔCt^ method. The data were analyzed using Applied Biosystem SDS (Ver. 1.4) software (Thermo Fisher Scientific, Milan, Italy).

### 4.4. Weighted Gene Co-Expression Network Analysis (WGCNA) and Module Identification

WGCNA is an approach utilizing gene expression data to construct co-expression networks weighted for high correlations [35] and was used in this study to evaluate the correlation between lncRNA and mRNA expression in NSCLC cell lines that are sensitive and resistant to EGFR-TKI. The gene expression dataset GSE80344 of sample biological duplicates (n = 16), publicly available at the NCBI’s Gene Expression Omnibus (GEO) database, was used as input for the WGCNA.

The gene expression raw microarray data were background corrected, log2 transformed, and quantile normalized using the *limma* package as previously reported [89]. Probes with low signal intensities and low variation were cut-off: 20,192 probes (mapping to 13,904 Ensembl gene ID) with signal intensities >5.5 and relative standard deviation (RSD) >0.3 were submitted to the following analysis: of these probes, 3591 map to lncRNAs (2386 Ensembl Gene ID).

The weighted correlation network analysis was performed with the WGCNA v.1.70.3 package of R v.4.1 software [35,90]. We used the WGCNA package to construct a weighted gene co-expression network, detect sub-clusters of genes, and correlate them to the phenotypic traits of our interest: intermediate EMT phenotype and ERL-resistance. Briefly, we obtained the expression correlation square matrix by calculating the Pearson correlation coefficient for all possible probe pairs. To construct a co-expression network, we used the scale-free topology criterion to choose the best parameter of the adjacency function: *a_ij_* = |*s_ij_*|*^β^* [90]. The fit was calculated using different *β* parameters (*soft threshold*) and selecting the *β* parameter of the fitting with R^2^ ≃ 0.9. The adjacency measures were transformed into topology overlapping matrix (TOM) scores (or weights), which reflect the relative interconnectedness between two nodes [91]. The TOM-based dissimilarity (*d^ω^* = 1 − *TOM*) was used for the hierarchical clustering of probes and the definition of the network *modules*. In particular, the modules (i.e., groups of highly correlated probes) were formed by hierarchical clustering (*method* = “average”), followed by the *Dynamic Tree Cut* algorithm (*deepSplit* = 2, *minModuleSize* = 30) and a merging of close clusters with the *Merge Dynamic* algorithm (*cutHeight* = 0.2).

### 4.5. Modules and Probes Selection

The first principal component of module gene expression data (module eigengene) was used to relate modules to each probe expression profile (module membership, MM) by the Pearson coefficient (*ρ*), and to phenotypical traits of cell lines by Spearman’s rank correlation coefficient (*ρ_s_*). The Spearman’s rank correlation was also used to calculate the gene/probe significance (GS), which in this work is the correlation between individual probe values and the biological trait of interest.

The student asymptotic *p*-values were calculated to assign a significance value to the correlation coefficients. *p*-values < 0.05 were considered significant.

For each gene/probe, connectivity (K) was represented by node degree (i.e., the sum of connection strengths with the other network nodes) and was calculated with three measures to study how connected a gene/probe is with respect to the other genes/probes: whole network connectivity (k), intramodular connectivity (kWithin), and scaled intramodular connectivity (kWithin scaled).

Modules of interest were selected according to their correlation with the EMT intermediate phenotype (“I vs. E”) and/or ERL-resistance (|*ρ_s_*| > 0.65, *p*-value < 0.05). Modules with a significant correlation (*p*-value < 0.05) with the “cell lineage” trait were discarded, as were modules with a not-significant correlation with the “I vs. M” trait (*p*-value > 0.05).

LncRNAs and mRNAs with significant MM and GS for intermediate EMT or erlotinib resistance greater than 0.7 were selected at probe level.

Module networks were finally imported into Cytoscape v.3.8.1 for visualization and topology analysis. For each node, we considered only the first 10 edges with a weight greater than the 0.8 quantile of module edge weights.

In the Discussion section, we summarized the gene parameters for module membership, gene significance, and connectivity as the average of probe values mapping to those genes.

### 4.6. Functional Annotations

Function annotations of modules were performed using the DAVID web tool [92] by using the official gene symbol identifiers and the terms of the BBID, BIOCARTA, KEGG, and Reactome databases. We performed pathway enrichment analysis, selecting terms with a count >2 and a *p*-value < 0.05.

## 5. Conclusions

In conclusion, in a NSCLC cellular model system, we identified three clusters of co-expressed genes whose expression correlates with resistance to EGFR-TKI and/or intermediate EMT phenotypes. These clusters of genes, with a potential biological significance for the selected NSCLC features, show strong links between some lncRNAs and mRNAs that could be a milestone for future mechanistic studies aimed at investigating the unknown function and action mechanisms of lncRNAs in NSCLC.

## Figures and Tables

**Figure 1 ijms-25-03863-f001:**
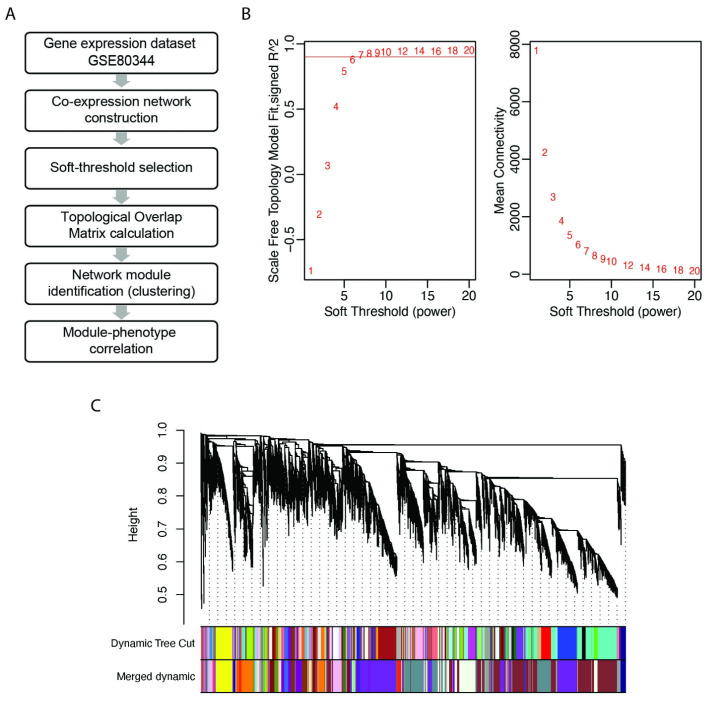
Weighted gene correlation network analysis (WGCNA) of NSCLC cell lines sensitive and resistant to EGFR-TKI (erlotinib) and distinct EMT phenotypes. (**A**) WGCNA pipeline. (**B**) Plots of scale-free topology fitting index (left) and mean connectivity (right) as a function of soft thresholds. Red horizontal line indicates the fit parameter we chose to evaluate models (R^2^ = 0.9). The soft threshold *β* = 7 is the first value above R^2^ = 0.9 with good mean connectivity. (**C**) Hierarchical clustering of probes according to their Topological Overlapping Matrix (TOM) dissimilarity score. The color bars show the assignment of modules performed by Dynamic Tree Cut and Merged Dynamic algorithms (see Methods section for details). Each branch represents genes; colored bars represent the modules containing a group of highly connected genes.

**Figure 2 ijms-25-03863-f002:**
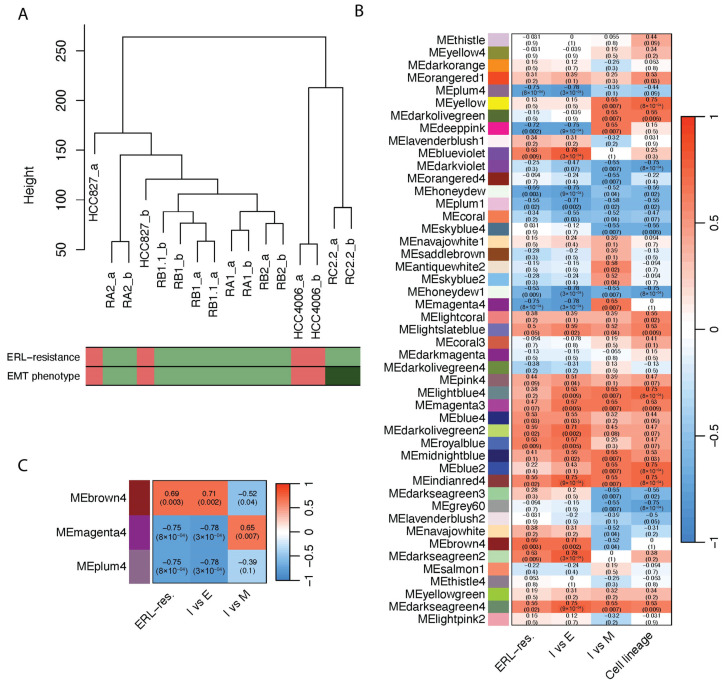
Identification of the key modules associated with ERL-resistance and intermediate EMT phenotypes. (**A**) Hierarchical clustering of NSCLC cell lines. Biological replicates show the highest degree of correlation within samples, represented by short vertical distances. Within each population, cell lines tend to cluster according to their biological features. Color bars indicate the phenotypes investigated: ERL-resistance indicates erlotinib-sensitive (red) and -resistant (green) cell lines; EMT phenotypes indicate epithelial (red), mesenchymal (dark green), and intermediate EMT (green) cell lines. (**B**) Heatmap of the module–trait relationships. Rows correspond to a module eigengene (ME), and columns correspond to the phenotypic traits. Each cell contains the corresponding correlation coefficient and *p*-value. The table is color-coded by correlation, according to the color gradient shown in the legend. (**C**) Heatmap of the module–trait relationships of selected modules. ERL-res = erlotinib resistance; *I* vs. *E* = intermediate EMT vs. epithelial cell lines; *I* vs. *M* = intermediate EMT vs. mesenchymal cell lines.

**Figure 3 ijms-25-03863-f003:**
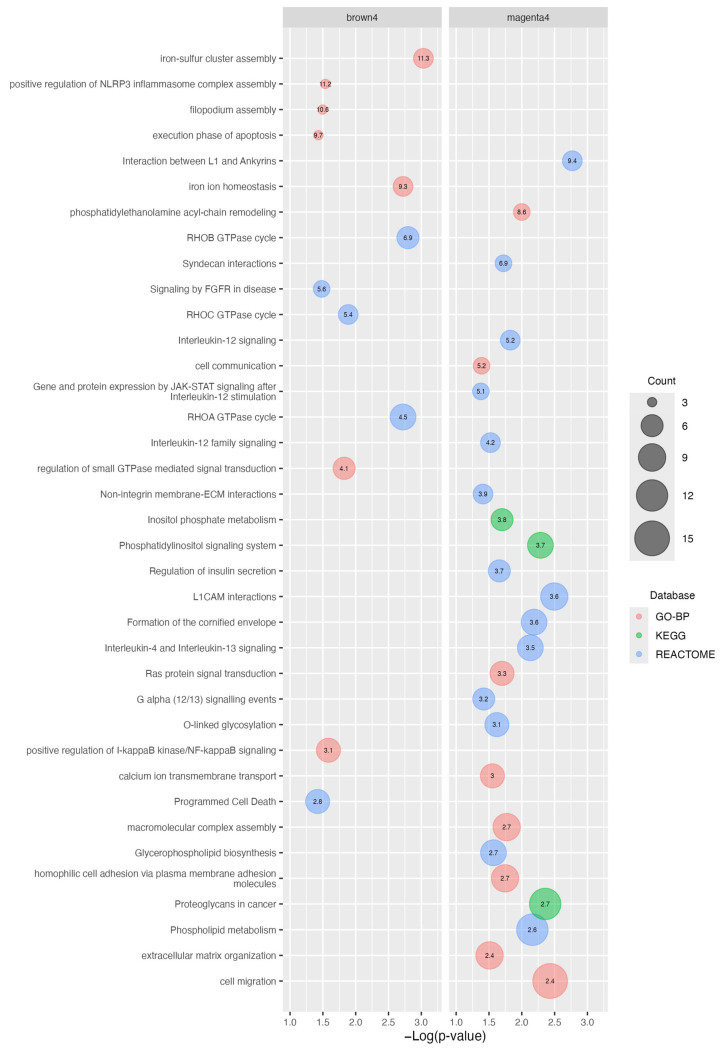
Functional enrichment analysis of the *brown4* and *magenta4* modules. Gene Ontology biological processes (GO-BP, in red), as well as KEGG (green) and Reactome (blue) enriched pathways, are shown in the plot. Terms are ranked on the X-axis according to the *p*-value and on the Y-axis according to the fold-enrichment (reported inside the bubbles). Bubble size is proportional to the count of module genes belonging to the term, as indicated in the legend.

**Figure 4 ijms-25-03863-f004:**
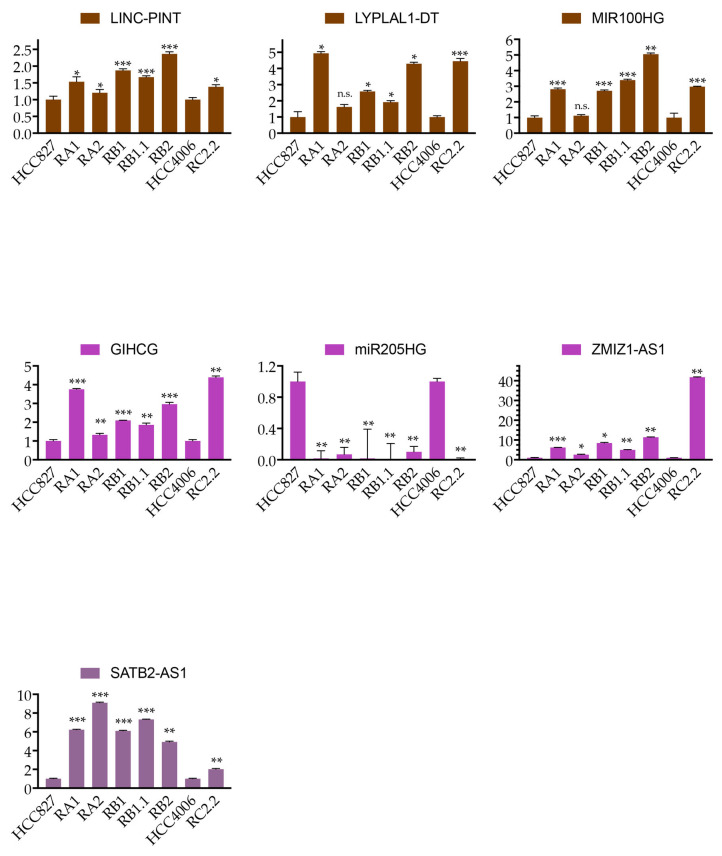
qPCR analysis of selected lncRNAs. qPCR analysis of the lncRNAs indicated is normalized to RPL31 mRNA and expressed relative to their levels in parental (HCC827 or HCC4006) cell lines (mean ± SD). qPCR data are representative of three independent experiments. Asterisks indicate significant *t*-test *p*-values in the comparison of a given derived cell line with the corresponding parental cell line: * *p* < 0.05; ** *p* < 0.01; *** *p* < 0.001; n.s. no significant *p*-values. Colors indicate the module membership of the selected lncRNAs, from the top: *brown4*, *magenta4*, and *plum4*.

**Figure 5 ijms-25-03863-f005:**
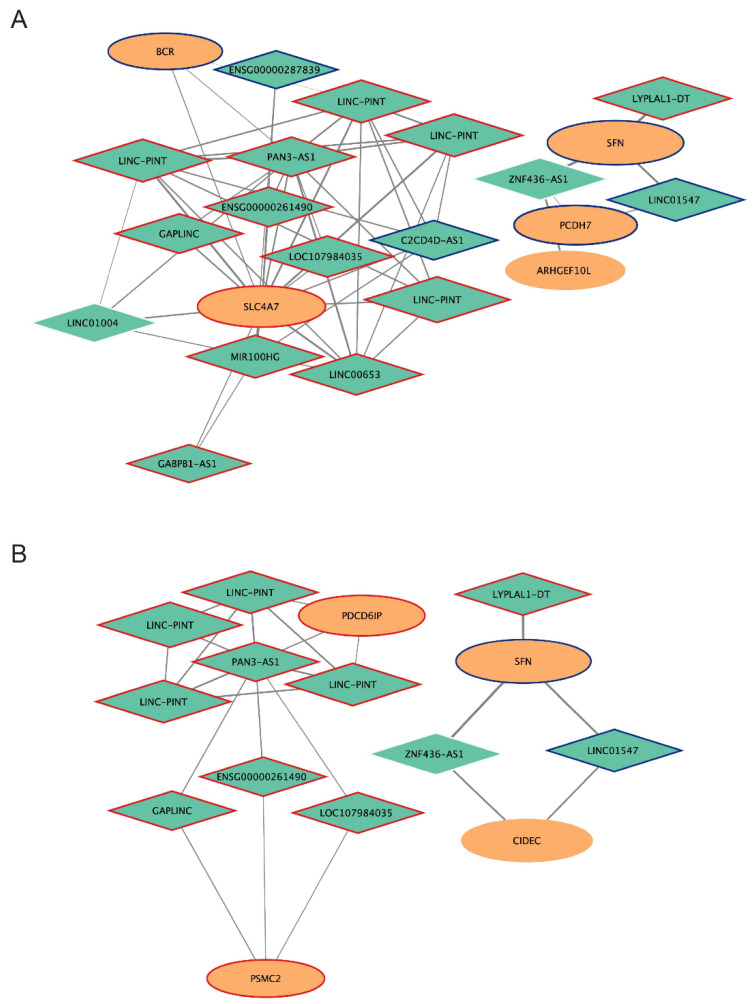
Subnetworks of the *brown4* module. Subnetwork of selected lncRNAs connected to mRNA of the *Rho GTPase* and/or *Rho GTPase regulation* pathways (**A**); apoptotic enriched processes and pathways (**B**) identified by the DAVID functional enrichment analysis of the *brown4* module. The selected lncRNAs (green diamonds) and mRNAs (orange ellipses) at probe-level are connected by edges whose sizes are proportional to their WGCNA weights: the higher, the thicker. Node size is proportional to the intramodular connectivity measure (kWhithinScaled), and node border color and width indicate the fold-change between Erl-resistant/intermediate EMT and -sensitive/epithelial NSCLC cell lines: red = upregulation (|fold-change| > 1.5); blue = downregulation (|fold-change| < 0.5); white = no expression variation; thickness = the higher, the thicker.

**Figure 6 ijms-25-03863-f006:**
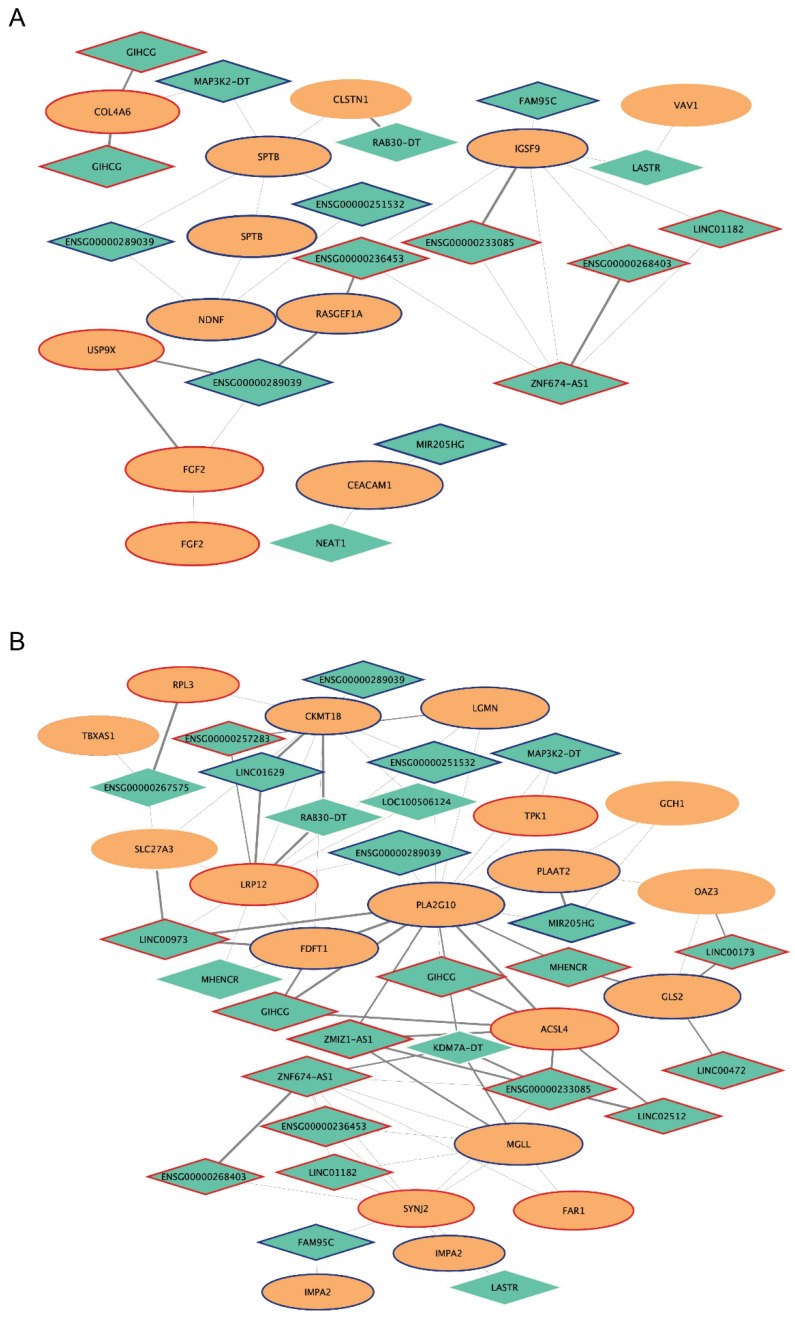
Subnetworks of the *magenta4* module. Subnetwork of selected lncRNAs connected to mRNAs of cell adhesion, cell migration, cell–extracellular matrix pathways (**A**); *metabolism*, and/or *lipid and phospholipid metabolism* (**B**) identified by the DAVID functional enrichment analysis of the *magenta4* module. The selected lncRNAs (green diamonds) and mRNAs (orange ellipses) at probe-level are connected by edges whose sizes are proportional to their WGCNA weights: the higher, the thicker. Node size is proportional to the intramodular connectivity measure (kWhithinScaled), and node border color and width indicate the fold-change between Erl-resistant/intermediate EMT and -sensitive/epithelial NSCLC cell lines: red = upregulation (|fold-change| > 1.5); blue = downregulation (|fold-change| < 0.5); white = no expression variation; thickness = the higher, the thicker.

**Figure 7 ijms-25-03863-f007:**
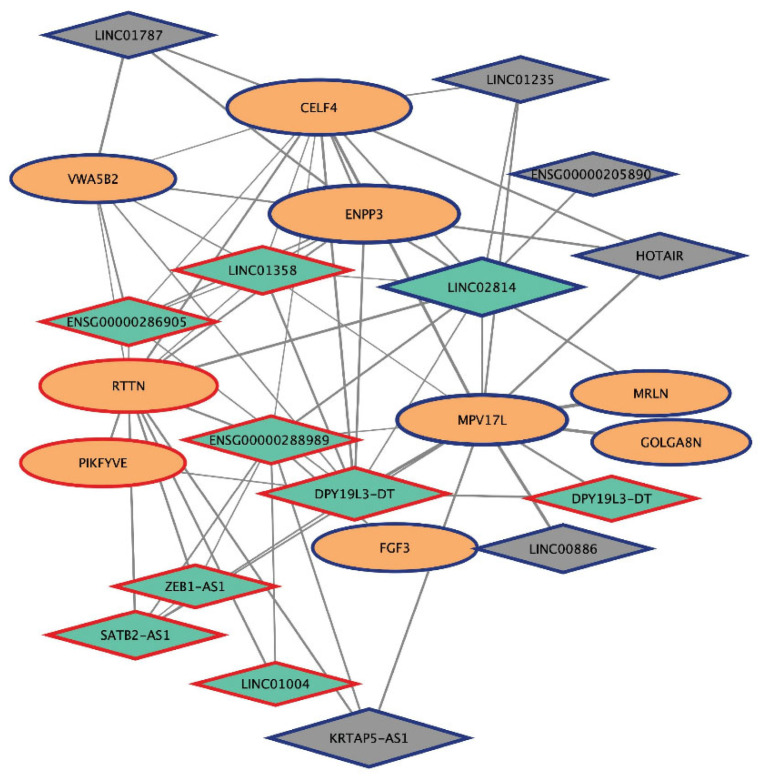
Network of the *plum4* module. lncRNAs (diamonds) and mRNAs (ellipses) at probe-level are shown in an organic layout. Nodes are connected by edges whose sizes are proportional to their WGCNA weights: the higher, the thicker. Node size is proportional to the intramodular connectivity measure (kWhithinScaled), and node border color and width indicate the fold-change between Erl-resistant/intermediate EMT and -sensitive/epithelial NSCLC cell lines: red = upregulation (|fold-change| > 1.5); blue = downregulation (|fold-change| < 0.5); white = no expression variation; thickness = the higher, the thicker. Nodes of selected lncRNAs are filled in green.

**Table 1 ijms-25-03863-t001:** Top lncRNAs of *brown4* and *magenta4* modules correlated to erlotinib resistance and intermediate EMT phenotypes.

Gene Name	Biotype	Cytoband Location	Module
*LINC-PINT*	lincRNA	7q32.3	*brown4*
*MIOS-DT*	divergent lncRNA	7p21.3	
*LYPLAL1-DT*	divergent lncRNA	1q41	
*LINC01547*	lincRNA	21q22.3	
*MIR100HG*	miRNA host gene	11q24.1	
*ZNF436-AS1*	antisense lncRNA	p36.12	
*LOC107984035*	uncharacterized	9p11.2	
*LINC00653*	lincRNA	20p11.23	
*LHX1-DT*	divergent lncRNA	17q12	
*GAPLINC*	lincRNA	18p11.31	
*PAN3-AS1*	antisense lncRNA	13q12.2	
*LINC01004*	lincRNA	7q22.3	
*ENSG00000261490*	uncharacterized	4p16.1	
*ENSG00000232850*	antisense lncRNA	9q34.11	
*C2CD4D-AS1*	antisense lncRNA	1q21.3	
*NFE2L1-DT*	divergent lncRNA	17q21.32	
*MIR4435-2HG*	miRNA host gene	2q13	
*ENSG00000234141*	lncRNA	7p21.3	
*ENSG00000287839*	uncharacterized	1q22	
*GABPB1-AS1*	antisense lncRNA	15q21.2	
*ENSG00000289039*	uncharacterized	10q26.3	*magenta4*
*MHENCR*	lincRNA	20q13.33	
*ENSG00000233085*	lincRNA	6q27	
*ENSG00000251532*	lincRNA	5p15.33	
*ENSG00000236453*	lincRNA	7q21.3	
*SPART-AS1*	antisense lncRNA	13q13.3	
*LINC00973*	lincRNA	3q12.1	
*MAP3K2-DT*	divergent lncRNA	2q14.3	
*ZNF674-AS1*	antisense lncRNA	Xp11.3	
*GIHCG*	lncRNA	12q14.1	
*MIR205HG*	miRNA host gene	1q32.2	
*ZMIZ1-AS1*	antisense lncRNA	10q22.3	
*NEAT1*	lincRNA	11q13.1	
*RAB30-DT*	divergent lncRNA	11q14.1	
*LINC01629*	lincRNA	14q24.3	
*LOC100506124*	lncRNA	2q24.3	
*ENSG00000267575*	lincRNA	19q11	
*LINC00472*	lincRNA	6q13	
*FAM95C*	lncRNA	9p13.1	
*LINC02512*	lincRNA	4q33	
*LINC01182*	lincRNA	4p15.33	
*LASTR*	lincRNA	10p15.1	
*ENSG00000268403*	antisense lncRNA	11p15.4	
*ENSG00000257283*	antisense lncRNA	12q22	
*KDM7A-DT*	divergent LncRNA	7q34	
*LINC00173*	lincRNA	12q24.22	

**Table 2 ijms-25-03863-t002:** Top lncRNAs of *plum4* module correlated to erlotinib resistance.

Gene Name	Biotype	Cytoband Location	Module
*LINC02814*	lincRNA	1q42.13	*plum4*
*DPY19L3-DT*	divergent lncRNA	19q13.11	
*ENSG00000288989*	uncharacterized	9q34.13	
*SATB2-AS1*	antisense lncRNA	2q33.1	
*LINC01358*	lincRNA	1p32.1	
*ENSG00000286905*	uncharacterized	2p25.3	
*ZEB1-AS1*	antisense lncRNA	10p11.22	
*LINC01004*	lincRNA	7q22.3	

## Data Availability

All data generated or analyzed during this study are included in this published article and its Appendix A.

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
