# Peer review of "Co-Expression Network Analysis Unveiled lncRNA-mRNA Links Correlated to Epidermal Growth Factor Receptor-Tyrosine Kinase Inhibitor Resistance and/or Intermediate Epithelial-to-Mesenchymal Transition Phenotypes in a Human Non-Small Cell Lung Cancer Cellular Model System"

_ijms, 2024, doi:10.3390/ijms25073863_

Round 1

Reviewer 1 Report

Comments and Suggestions for Authors

Please, see the PDF.

Manuscript ID: ijms-2894243

Title: Co-expression networks analysis unveiled lncRNA-mRNA links correlated to EGFR-TKI resistance and/or intermediate epithelial to mesenchymal transition phenotypes in a human NSCLC cellular model system

The following corrections need to be done by the authors:

    I appreciate the thorough exploration the authors have undertaken in their study. The strength of the study lies in the rigorous methodology followed, the careful exploration of LncRNA-mRNA correlation patterns and their presentation in the study with complete supplementary information. While the manuscript is strong overall, there are few points that can be considered to improve the overall presentation of the results.

1.       In line no. 62, “Importantly, EMT is a complex dynamic and not binary process characterized by a spectrum of intermediate phenotypes” can be changed to “Importantly, EMT is a complex, dynamic and multifaceted process characterized by a spectrum of intermediate phenotypes”

2.       In line no 136, “we focused our attention on modules with high significant correlation with ERL-res and “I vs E” traits, and significantly correlated with the “I vs M” trait.” Does that mean to have significant correlation to ERL-res, I Vs E and I vs M phenotypes? If so, the line can be reframed accordingly.

3.      At line no. 109, the citation for WGCNA is incorrect. “The WGCNA analysis was performed in R complying with the pipeline defined by Peter Langfelder & Steve Horvath (Figure 1A) [32].” Reference No. 32 refers to “Mathematical Modeling of Plasticity and Heterogeneity in EMT.”. It needs to be corrected.

4.       In line no. 131, “For the definition of the intermediate EMT phenotypes we used two comparisons:  “I vs E” “, to select modules with gene expression differences between cell lines with intermediate EMT phenotypes (I) and epithelial cell lines (E); and “I vs M”, to select modules with gene expression differences between cell lines with intermediate EMT phenotypes and mesenchymal cell lines (M). “

As the authors mention I Vs E, is that the samples are given binary codes for Intermediate and Epithelial cell lines or the authors have calculated gene expression differences between the samples and used it for calculating the significance needs more clarity. While there are only 2 samples for Mesenchymal cell line model, as inferred from Figure 2A, the authors can provide a separate sample information table in supplementary including the Mesenchymal, Intermediate or Epithelial phenotype information of the cell lines.

5.      In line no. 145, it is mentioned that “Next, to verify that connectivity and memberships of individual members of modules had good correlations with our phenotypic trait of interest, we performed a correlation analysis of module membership and gene significance of probes as well as gene significance and intramodular connectivity (Figure S1, Table S1). “

So, the correlation analysis of (i) Module membership against gene significance and (ii) Intramodular connectivity against gene significance was conducted to verify whether the connectivity and memberships of individual module members show strong associations to the phenotype of our interest. In the analysis performed, if the authors have used the absolute values of Module membership or gene significance values, it should be mentioned.

In the second figure of the manuscript Figure 2C, it shows brown4 module to be positively associated to ERL-resistance, whereas the other two are negatively correlated. Considering that figure, the genes with higher module memberships in magenta4 and plum4 module are expected to have negative correlation to ERL-resistance. So, Figure S1, is expected to have negative correlations to ERL-resistance, which is not seen. This point needs to be clarified by the authors.

6.      In Line no. 213. “LINC-PINT, LYPLAL1-DT and MIR100HG of the brown4 module; GIHCG, ZMIZ1-AS1 of the magenta4 module and SATB2-AS1 of the plum4 module are all upregulated in the erlotinib resistant cell lines, while MIR205HG of the magenta4 module is downregulated (Figure 4).”

This indicates that while certain LncRNAs exhibit upregulation in erlotinib-resistant cell lines others show downregulation. Though some of this information are provided in the supplementary Table S3, it is advisable to mention it clearly the manuscript that, in Table S3 the direction of association of lncRNAs to erlotinib resistance is mentioned. This will help readers for the better understanding of manuscript.

7.      The resolution of Figures 5, 6 and 7 should be increased. Additionally, consider increasing the font size for improved visibility.

8.      In the Line no. 301, it is mentioned that “Erl-sensitive/epithelial and resistant/intermediate EMT NSCLC cell lines: red = up-regulation (|fold-change| > 1.5); blue = down-regulation (|fold-change| < 0.5); white = no expression variation; thickness= the higher, the thicker.” If the fold difference is calculated for ERL-resistant/intermediate Vs ERL-sensitive/epithelial then the order of representation should be changed. Currently it is like Sensitive Vs resistant from the above lines of figure legend.

9.      In the Figure 5A, there are several LINC-PINT LncRNAs found connected to each other as well as to the mRNAs in the network. As multiple probes can represent a single gene / LncRNA in this study, it can be mentioned clearly in the figure 5 discussion.

Reviewer 2 Report

Comments and Suggestions for Authors

Authors investigated mRNA-lncRNA co-expression patterns in a cellular model system of non-small cell lung cancer sensitive and resistant to the EGFR tyrosine kinase inhibitors erlotinib/gefitinib. Genome-wide RNA expression was quantified for weighted gene co-expression network analysis (WGCNA) to correlate the expression levels of mRNAs and lncRNAs. Aim to unveil insights into the complex mechanisms of non-small cell lung cancer targeted therapy resistance and epithelial to mesenchymal transition (EMT). The experimental design is reasonable and interesting.

Comments and questions:

1.     For section 2.1, 2.2 and 2.3, the result and is too short or no result description at all.

2.     Supplement relevant hypothesis or conclusion after each result description.    

3.     The resolution of Figure 3, Figure 5, Figure 6, and Figure 7 is too low, I can barely see the inside number or word.

In summary, I recommend this paper to be minorly revised before accepting.
